# Superwavelength self-healing of spoof surface sonic Airy-Talbot waves

Hao-xiang Li[1,2,4], Jing-jing Liu [1,4], Zhao-xian Chen [1], Kai Wu[1], Bin Liang [1] ✉, Jing Yang [1] ✉, Jian-chun Cheng [1] ✉ & Johan Christensen [3] ✉

Self-imaging phenomena for nonperiodic waves along a parabolic trajectory encompass both the Talbot effect and the accelerating Airy beams. Beyond the ability to guide waves along a bent trajectory, the self-imaging component offers invaluable advantages to lensless imaging comprising periodic repetition of planar field distributions. In order to circumvent thermoviscous and diffraction effects, we structure subwavelength resonators in an acoustically impenetrable surface supporting spoof surface acoustic waves (SSAWs) to provide highly confined Airy-Talbot effect, extending Talbot distances along the propagation path and compressing subwavelength lobes in the perpendicular direction. From a linear array of loudspeakers, we judiciously control the amplitude and phase of the SSAWs above the structured surface and quantitatively evaluate the self-healing performance of the Airy-Talbot effect by demonstrating how the distinctive scattering patterns remain largely unaffected against superwavelength obstacles. Furthermore, we introduce a new mechanism utilizing subwavelength Airy beam as a coding/decoding degree of freedom for acoustic communication with high information density comprising robust transport of encoded signals.

Artificially structured media in materials and physics sciences have become a landmark in the scenery of man-made wave responses comprising unusual acoustic, optical and mechanical characteristics[1–10]. Negative refraction[11–15], cloaks of invisibility and unhearability[16–20], and topological wave-based phases[21–26] are a few of many exciting directions in the seemingly ever-growing frontier. These exceptional responses and wave functionalities stem from diffractive characteristics at superwavelength scales but are also engineered using units that are built at subwavelength dimensions.

The Talbot effect is a classical phenomenon in optics discovered in 1836[27] in which near-field diffraction of a periodic structure replicates itself at a designated position referred to as the Talbot distance[28–34]. Another prominent diffraction phenomenon is the Airy beam which displays a parabolic intensity bending of its main lobe with exceptional resilience against perturbations. This effect that was

named after Sir George Biddel Airy, who derived the Airy integral in connection with optical caustics, has demonstrated its significance through the ability to direct light or sound along curved paths[35–38]. Such unusual yet spectacular bent routes for waves have particularly capitalized on the use of metamaterials and metasurfaces, which enable the necessary phase and amplitude textures for the desired outcome[39–41].

The Airy-Talbot effect, which is reminiscent, yet quite different from the Talbot effect, depicts that a superposition of fundamentally accelerating beams can reproduce themselves at fixed intervals along curved trajectories, which greatly broadens the potential of conventional self-imaging. Although previously reported studies have explored the Airy-Talbot effect in optics[42–45], the observation of the Airy-Talbot effect for acoustic waves still remains highly challenging, which is chiefly ascribed to the following reasons. First, caused by

[1]Key Laboratory of Modern Acoustics, MOE, Institute of Acoustics, Department of Physics, Collaborative Innovation Center of Advanced Microstructures, Nanjing University, Nanjing 210093, P. R. China. [2]College of Information Science and Technology, Nanjing Forestry University, Nanjing 210037, P. R. China. [3]IMDEA Materials Institute, Calle Eric Kandel, 2, 28906 Getafe, Madrid, Spain. [4]These authors contributed equally: Hao-xiang Li, Jing-jing Liu. ✉e-mail: liangbin@nju.edu.cn; yangj@nju.edu.cn; jccheng@nju.edu.cn; johan.christensen@imdea.org

severe diffraction in free space, an acoustic Airy beam undergoes rapid spreading during propagation, which hinders the collective superposition of several Airy beams and perfect periodic reproduction of the subwavelength amplitude profiles. Moreover, the attenuation coefficient of sound waves in free space is significantly larger than that of optical waves, which means that thermoviscous losses restrict the extended propagation lengths and the formation of self-imaging in the acoustic regime. Lastly, considering the long wavelength of acoustic waves, the experimental implementation will be unpractically large in comparison to the optical counterpart[46]. Conclusively, the experimental observation of the acoustic Airy-Talbot effect still remains challenging.

In this article, we report the theoretical and experimental realization of the Airy-Talbot effect using sound waves. By employing a metasurface, we combine the Talbot effect with Airy beams and quantitatively evaluate their resilience against superwavelength rigid objects, during which sonic self-healing persists. In order to overcome the above-named problems in terms of attenuation and diffraction, we structure Helmholtz resonators (HRs) into an otherwise acoustically impenetrable surface to sustain SSAWs. Our demonstration showcases a significant reduction in the transverse lobe profiles, alongside the capacity to extend the self-imaging of multiple Airy beams along

curved paths to farther distances, both thanks to the subwavelength SSAWs. Thus, the realization of surface-confined self-imaging along curved trajectories at deeply subwavelength scales facilitates distinct self-healing capabilities against superwavelength objects. Furthermore, we experimentally exemplify resilient parallel multiplexing of acoustic images by utilizing subwavelength Airy beam as a novel encoding/decoding degree of freedom in two dimensions. Despite the reduction of system dimensions in comparison with the existing free-space sound communication paradigm, our proposed scheme enables high-quality robust data transmission, which should be significant for diverse applications ranging from on-chip signal transfer[47] and filtering to acoustofluidic manipulation[48].

## Results

### Theory of spoof surface sonic Airy-Talbot waves

Figure 1a schematically depicts the robust transmission mechanism of the Airy-Talbot effect. The Airy beams, which are scattered by the superwavelength cylinder located on the transmission path, resume their amplitude envelopes at regular distances. A line array of loudspeakers is utilized to emit the accelerated wave packet of almost arbitrary shape. The holey metasurface consisting of HRs, functions as the acoustic guiding surface, which sustains the propagation of SSAWs

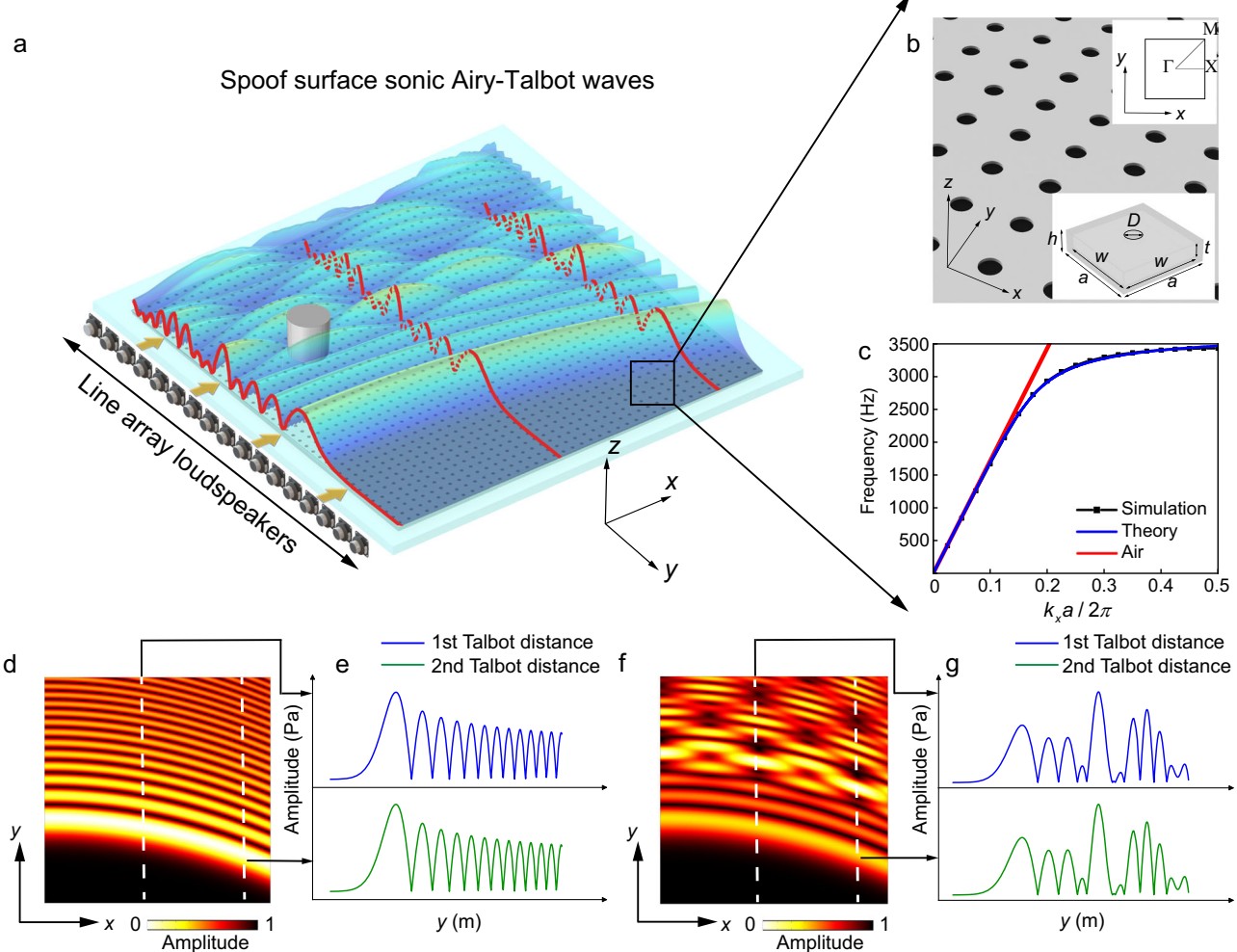

**Fig. 1 | Schematic of superwavelength self-healing effect of spoof surface sonic Airy-Talbot waves. a** The metasurface consisting of Helmholtz resonators (HRs) above which the spoof surface sonic Airy-Talbot waves are launched. The red line denotes the distribution of the incident sound waves' amplitude which resumes to its original shape at regular distances. The image of the speakers is created with Adobe Photoshop 2022. **b** Zoomed-in view of the metasurface shown in (**a**). The lower-right inset shows the unit cell and the upper-right inset shows the corresponding first Brillouin zone. **c** Dispersion curve of the metasurface, where the symbols, blue and red lines represent the theoretical, numerical results and air line, respectively. Pressure amplitude plots and profiles at two self-imaging planes of transmitting **d**, **e** one and **f**, **g** two Airy beams, respectively.

(as shown in Supplementary Fig. 1). A microphone is located at the self-reproducing plane to detect the transmitted signals.

As depicted in Fig. 1b, our metasurface under study contains periodically arranged Helmholtz-like subwavelength resonators, whose circular necks are in plane with the surface. The lattice constant $a$ is 2 cm, the diameter of the aperture $D$ is 0.7 cm, the depth of the resonator $h$ is 0.75 cm, the length of the cavity $w$ is 1.8 cm, the depth of the cavity $t$ is 0.55 cm, and the wall thickness $d$ is 0.1 cm, respectively.

For the lossless metasurface in air, ($\rho_0 = 1.21$ kg/m³, $c_0 = 343$ m/s), we obtain the SSAW dispersion relation as follows[49–51]

$$k_x = \frac{\omega}{c_0} \sqrt{1 + \frac{\omega^2}{(\omega_{HR}^2 - \omega^2)^2} \left(\frac{\rho_0 c_0}{M_{HR} a^2}\right)^2}, \qquad (1)$$

where $k_x$ is the wavenumber, $\omega$ is the angular frequency, $\omega_{HR} = \sqrt{1/M_{HR} C_{HR}}$ is the resonance frequency of the HRs. Since the calculated equi-frequency contour (EFC) at the working frequency is near perfectly circular, we only need to calculate the dispersion relation along $k_x$ (see Supplementary Note 2 for more details). $M_{HR}$ and $C_{HR}$ are the acoustic mass and the acoustic capacitance of HRs, respectively, which can be written through the structural parameters

$$M_{HR} = \frac{4\rho_0 d}{\pi D^2}, \quad C_{HR} = \frac{t(a-2d)^2}{\rho_0 c_0^2}. \qquad (2)$$

Figure 1c displays the computed SSAW dispersion relation, in which the analytical expression from Eq. (1) displays a remarkably good agreement with finite element simulations. When cutoff is reached, i.e., $\omega = \omega_{HR}$, the band asymptotically approaches a perfectly flat band for $k_x \to \infty$. This means that even when additional Airy beams are superimposed, we are able to reduce the transverse extent, i.e., generating the entire spatially bent Airy beam profile at compact source array scales. To further demonstrate the subwavelength propagation property, we compare the amplitude profiles of a single Airy beam propagating in free space and above the metasurface, showing the subwavelength spatial compression above the SSAWs metasurface (see Supplementary Notes 3–5 for more details). Additionally, along the longitudinal axis, the band flattening effect shrinks the effective wavelength, which in turn extends the Talbot distance along the propagation length, enabling Airy beams to interfere adequately at pronounced distances (see Supplementary Note 6 wherein we demonstrate the capability of metasurfaces in modulating Talbot distances).

In the next part, we prove mathematically the acoustic Airy-Talbot effect. In homogeneous isotropic discrete media, the Airy beam can be written as

$$\phi(x,y) = c_n \text{Ai} \left[ y - \left(\frac{x}{2}\right)^2 \right] \exp \left[ i \left(\frac{xy}{2}\right) - i \frac{x^3}{12} \right], \qquad (3)$$

where the $c_n$ is the arbitrary coefficient, Ai is the Airy function, $x$ and $y$ are the normalized propagation distance and transverse coordinate, $i$ is the imaginary unit. Figure 1e depicts the simulated results of amplitude distributions along two self-imaging planes marked by dashed lines in Fig. 1d. Throughout the paper, the finite element method based on COMSOL Multiphysics software is used for the numerical simulations.

Furthermore, we superpose the Airy beams at periodic intervals and write the sound field as

$$\phi(x,y) = \left\{ \sum_n c_n \text{Ai} \left[ y - \left(\frac{x}{2}\right)^2 - n\Delta \right] \exp \left( -\frac{1}{2} i n \Delta x \right) \right\} \exp \left[ i \left(\frac{xy}{2}\right) - i \frac{x^3}{12} \right], \quad (4)$$

where $\Delta$ is the interval of Airy beams. The Talbot length can be deduced as (see Supplementary Note 7 for detailed derivations)

$$x_T = \frac{4\pi}{\Delta}. \qquad (5)$$

The coordinate variables $x$, $y$ and the interval period of Airy beams $\Delta$ in Eqs. (3–5) are dimensionless. For simplicity but without losing generality, we use two Airy beams to display the Airy-Talbot effect and set the interval between each two Airy beams as 0.45 m to obtain a suitable self-imaging distance (Fig. 1f, g). It is worth noting that more incident Airy beams will allow for a better Talbot effect, which, however, calls for a larger experimental setup and a more complicated design of the sound source. In our current design, to facilitate the experimental implementation, we emit two Airy beams to display the Airy-Talbot effect, which is sufficient to validate the mechanism of the self-imaging effect (See Supplementary Fig. 9 for more details). In addition, for the purpose of avoiding thermal viscosity and anisotropy, we set the operation frequency at 3170 Hz, which is slightly off-resonance (see Supplementary Note 9 for the discussion of thermo-viscous losses). The sound velocity of the SSAWs is calculated as $c = 253.6$ m/s. The Airy-Talbot effect is mathematically proved by the calculated similarity between the amplitude profile at the sound source with that at the self-producing plane where $x = nx_T = n \times 0.65$ m ($n = 1, 2$), as demonstrated in Supplementary Note 7.

### Experimental demonstration of the self-reconstruction and self-healing effect

Considering the scattering of sound waves, the presence of obstacles usually has an adverse impact, which is the main drawback in areas such as communications. With the unique feature of self-healing[35,36,52,53], Airy beams can remain largely intact against superwavelength scatterers. Here, we demonstrate experimentally the Airy-Talbot effect, wherein the accelerated wave packet remains resilient in the presence of superwavelength obstacles along its trajectory, ensuring a notable level of transmission efficiency.

Figure 2a shows the experimental setup and structured surface sample fabricated by 3D printing. The measurement is carried out in an anechoic chamber to eliminate the undesired reflections from the boundaries. Solid material for the SSAW device is chosen as acrylonitrile-butadiene-styrene (ABS), which can be considered acoustically rigid. The surrounding columns of holes are filled with sound-absorption materials to mimic a perfect absorption boundary for the SSAWs. The entire sound field near the metasurface sample is measured utilizing a 1/4-in. free-field microphone (Brüel & Kæjr type-4961) which is attached to a 3D stepping motor to scan the target region point by point. Using full-wave simulations, a complex sound source, whose amplitude and phase are asymmetric along the $y$-axis and modulated as denoted by blue solid lines in Fig. 2b, c respectively, is utilized to effectively mimic the incident SSAWs. Experimentally, an array of 26 1.3-in. loudspeakers with a diameter of 3.3 cm is used for such a system, whose amplitude profile is shown in Fig. 2b. Figure 2b, c illustrates the discrete amplitudes and phases of each channel, as indicated by the solid lines in red and blue. A single-chip-microcomputer Arduino Mega 2560 is utilized to send the appropriate signals to each loudspeaker. Considering the size of our source, the acoustic field can only be reproduced a limited number of times. Yet, extending the source length increases the number of reproductions (see Supplementary Note 10 for more discussions about the influence of the source's length on the Airy-Talbot effect).

In order to explore quantitatively the influence of the size of an obstacle on the Airy-Talbot effect, cylinders with radii of 1.3 cm, 5.1 cm, 11.2 cm, 12.7 cm (correspond to $ka = 1, 4, 8.8, 10$) are successively placed ($x = 0.15$ m, $y = 0.6$ m) in the acoustic path of the main lobe of the Airy beams, as shown in Fig. 2d–g.

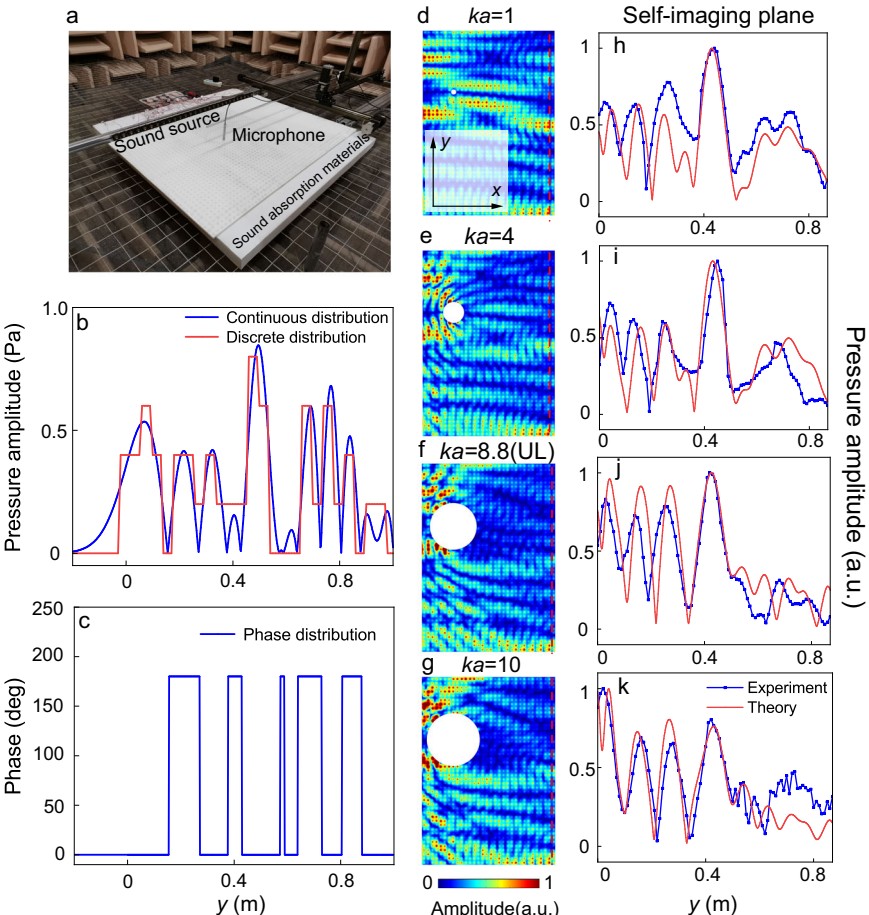

**Fig. 2 | Design of the acoustic SSAW metasurface and self-reconstruction of the Airy-Talbot effect. a** Experimental setup and device model. **b** Amplitude distribution of the source comprising continuous (blue) and discretized (red) curves. **c** Phase distribution. **d–g** Simulated acoustic fields with cylinders ($ka$ = 1, 4, 8.8 (upper limit), 10) placed at ($x$ = 0.15 m, $y$ = 0.6 m) above the designed SSAW device. $k$ is the equivalent wavenumber and $a$ is the radius of the obstacle. **h–k** Amplitude profiles along the dashed lines denoted in (**d–g**). The theoretical and experimental results are denoted by the red and blue lines, respectively.

We find that the main lobe that intentionally is being obstructed, as expected, leaves an acoustic shadow behind the obstacle. However, following the bent trajectory of the Airy-Talbot beam, it is visible that self-healing takes place in that energy flux around the obstacle gradually converges into the center to resuscitate the main lobe. To unambiguously demonstrate this important feature, we draw the amplitude profiles along the self-imaging plane in Fig. 2h–k. Similar to the input and output signals without the object marked by the gray and pink dashed lines, the diffractive fingerprints of the Airy-Talbot effect are shown by the pronounced pressure peak of the main lobe at the center of the self-imaging plane ($y$ = 0.43 m). Here, we find that $ka$ = 8.8 marks the upper limit (UL) at which self-healing prevails, i.e., where the said peak remains almost unchanged and robust transmission is guaranteed. Once $ka$ exceeds the upper limit, the peak value will gradually decrease and the amplitude profile will no longer be consistent with the sound source's (see Supplementary Notes 11 and 12 where we demonstrate transmission robustness with scatterers of different radii by measuring the acoustic field).

**Two-dimensional spatial multiplexing via evanescent acoustic Airy-Talbot modulation**

The afore-discussed capability of simultaneously transmitting multiple signals along a curvy trajectory, suggests a possibility to encode information using parallel data transfer. In the following, we experimentally showcase the application of the Airy-Talbot effect for the realization of a two-path parallel communication system. A cylinder with a radius of 11.2 cm ($ka$ = 8.8) is placed at ($x$ = 0.15 m, $y$ = 0.6 m), with the intent to deliberately obstruct the data transmission of channel 1 (Ch1). We also prove the effectiveness when both channels are blocked (see Supplementary Fig. 13). The experimentally executed Airy-Talbot communication is shown in Fig. 3a, b, which clearly verifies its resilience in multiplexed data. For simplicity, we input binary images that only contain two types of pixels "1" and "0" encoded as transmitted wave with normalized amplitude of 1 and 0 respectively. By switching between Airy-Talbot and ordinary Airy beams in the time domain, the emitting source is modulated to generate the multiplexed data which is characterized by multiple binary signals transmitting along different channels parallelly. To be specific, a single Airy beam launching produces detectable signals in only one channel (corresponding to {1,0} or {0,1}), while the Airy-Talbot mode displays simultaneously two peaks resulting from the interference of two Airy beams ({1,1}) (see Supplementary Notes 14 and 15 for the method of parallel transmission and extracting the data stream). The important self-healing effect ensures the restoration of the wavefront carrying information along its path of Ch1 (Fig. 3c, d), which results in high accuracy in data transmission and perfect reconstruction of encoded images despite the presence of the superwavelength obstacle. It is worthwhile that our proposed scheme is fundamentally different from the acoustic spatial multiplexing which is interesting in underwater communication[54] and commonly relies on the use of vortex beams in three-dimensional systems[55], as it utilizes subwavelength Airy beam as a new encoding/decoding degree of freedom for acoustic

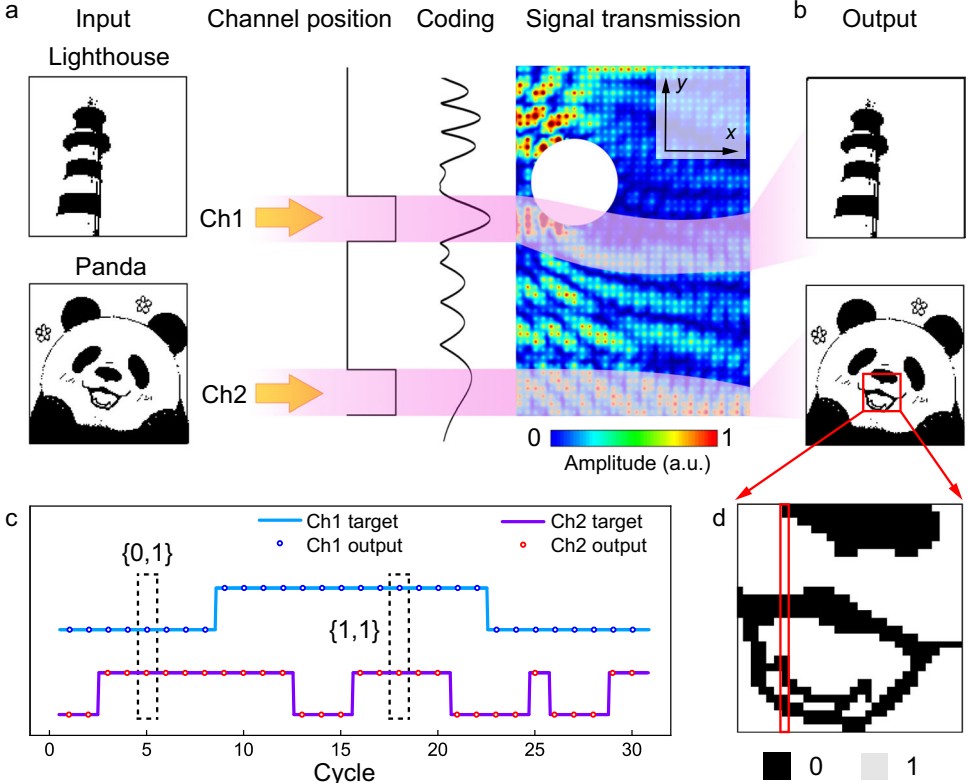

**Fig. 3 | Parallel transmission of two binary images via Airy-Talbot modulation.** **a** Schematic diagram of the digital coding and parallel transmission using Airy-Talbot sound. Two binary images of a lighthouse and a panda (each with 128 × 128 pixels) using two channels (Ch1/Ch2) are encoded by Airy beams and then transmitted along curved paths marked in purple. Ch1 is obstructed by a superwavelength ($ka = 8.8$) cylinder. **b** Images are independently retrieved from the received data carried by Airy beams in Ch1 and Ch2. **c** Comparison between the objective and received output. **d** Zoomed-in view of the red box part in (**b**). The two colors of pixel are encoded by binary information "0" and "1". The images of lighthouse and panda are hand-painted and converted from RGB color to binary images in Adobe Photoshop 2022.

communication with high information density in two dimensions and enables the reduction on the system dimension while keeping a high-quality robust acoustic data transmission.

## Discussion

In conclusion, the Airy-Talbot effect is experimentally demonstrated for sound on a metasurface. Thanks to the man-made dispersion of the SSAWs that carry the Airy-Talbot beams, one is able to engineer the structural parameters of the metasurface towards deeply subwavelength scales. Different from the conventional Talbot effect, the spatial superposition of Airy functions makes the self-imaging of aperiodic signals possible[56]. In addition, inspired by the Talbot effect's applications, the Airy beams could be introduced to conventional spatial multiplexing techniques, offering the possibility to further enhance communication efficiency. Furthermore, our mechanism is general and enables the design of more versatile functional devices in other systems. It is worth mentioning that in addition to the acoustic trapping and micromanipulation which Airy beams have been already used for[57-59], our demonstration, comprising Airy beams with self-healing features, is ideal in the use for non-topological resilient data transfer of parallel binary signals and may have far-reaching implications in diverse fields ranging from high-capacity communication to on-chip applications.

## Methods
### Numerical simulation
Throughout the paper, the simulations are conducted with a finite element method based on COMSOL Multiphysics software. The physical properties of air at 293.15 K and standard atmospheric pressure are mass density $\rho_0 = 1.21$ kg/m$^3$, sound speed $c_0 = 343$ m/s, thermal conductivity $\kappa = 0.0258$ W/(m·K), dynamic viscosity $\mu = 1.81 \times 10^{-5}$ kg/(m·s), specific heat at constant pressure $C_p = 1.005 \times 10^3$ J/(kg·K) and ratio of specific heats $\gamma = 1.4$. The solid material for the SSAW device is chosen as acrylonitrile-butadiene-styrene (ABS) with density 1180 kg/m$^3$ and sound speed 2700 m/s.

### Experimental configuration
The experiment is performed in an anechoic chamber without interference from other obstacles. The sample of SSAW metasurface is fabricated via 3D printing technology. Twenty-six loudspeakers (HiVi, model B1S) are linearly arranged on an acrylic glass plate and a single-chip microcomputer Arduino Mega 2560 is used to control the amplitude and phase of the drive signal of each loudspeaker. One 1/4-in. free-field microphone (Brüel & Kæjr type-4961) is placed at the self-imaging plane to detect the signals. The real-time signals are recorded with Brüel & Kæjr PULSE 3160-A-042 2-channel analyzer.

## Data availability
The data that support the findings of this study are available from the corresponding author upon request.

## Code availability
The code that supports the findings of this study is available from the corresponding author upon request.

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

## Acknowledgements
This work was supported by the National Key R&D Program of China (Grant No. 2022YFA1404402 to B.L.), the National Natural Science Foundation of China (Grant Nos. 11634006 to J.Cheng, 12174190 to J.Y., 12304493 to J. L.), Jiangsu Provincial Natural Science Foundation of China (Grant No. BK 20230767 to J. L.), High-Performance Computing Center of Collaborative Innovation Center of Advanced Microstructures and a Project Funded by the Priority Academic Program Development of Jiangsu Higher Education Institutions. J.Christensen acknowledges support from the Spanish Ministry of Science and Innovation through a Consolidación Investigadora grant (CNS2022-135706).

## Author contributions
H.L. and J.L. performed the theoretical simulations; H.L., J.L., Z.C. and K.W. designed and carried out the experiments; H.L., J.L., B.L., J.Y. and J.Christensen wrote the paper; H.L., B.L., J.Y., J.Cheng and J.Christensen guided the research. All authors contributed to data analysis and discussions.

## Competing interests
The authors declare no competing interests.
