## [Peer Review File · Nature Communications]

Superwavelength self-healing of spoof surface sonic Airy-Talbot wavesReviewer #1 (Remarks to the Author):

The authors combine Talbot effect with Airy beam to show that sound waves can remain largely unaffected against large obstacles in the propagating path. The Airy-Talbot waves take full advantage of the self-healing property of Airy beam and the self-imaging property of Talbot effect. This work represents a very good example of how physicists cleverly employ metamaterial-based SSAW that controls sound at deep-subwavelength scale to enable such an important effect. Moreover, they harness the distinct property of Airy-Talbot waves to realize interesting application of multiplexing communication with great potential. All these functionalities are experimentally demonstrated, and the results are well presented. I will recommend this work to be published in Nature Communications if the following issues are well addressed:

- 1) All the phenomena are studied in holey metasurface that sustains SSAWs. A plenty of simulations are conducted to compare the Airy-Talbot effect in this circumstance with that in air. Some detailed discussion should be given to explain why the metasurface holds better performances.
- 2) In Fig.2, the simulated acoustic fields for different cylinders are well presented. For the measured results, only the line plots are given. The according acoustic field distributions should also be measured to make the results more convincing.
- 3) The self-imaging of the incident source should be reproduced at the periodic intervals of Talbot length, so I wonder what's the acoustic field distribution at the next Talbot length.
- 4) In my opinion, more incident Airy beams will allow a better Talbot effect and more communication channels, but why only two Airy beams are used in this work?

Reviewer #2 (Remarks to the Author):

In this manuscript, the authors study the self-imaging of waves for nonperiodic waves along a parabolic trajectory encompassing both the Talbot effect and accelerating Airy beams. Beyond the ability to guide waves along a bent trajectory, the self-imaging component adds invaluable advantages towards lensless imaging comprising periodic repetition of planar field distributions. In order to circumvent adverse effect of viscothermal losses and diffractive spread, they structure subwavelength resonators in an acoustically impenetrable surface sustaining spoof surface acoustic waves (SSAWs) to provide highly confined Airy-Talbot effect. Moreover, these spoof surface sonic Airy-Talbot beams that entirely are controlled by geometrical means, enable both extended Talbot distances and compressed subwavelength lobes in the perpendicular direction. From a linear array of loudspeakers, we judiciously phase-and-amplitude launch sound above the structured surface and demonstrate how the scattered fingerprints of the Airy-Talbot effect remain largely unaffected against superwavelength obstacles, ideal for acoustic functional devices enabling robust transport of encoded signals. The topic is interesting. However, some questions and details should be noted and modified by the author.

1. What is the viscothermal losses?
 2. What is the difference between the acoustic Airy-Talbot waves and the Airy-Talbot plasmon?
 3. What is the physical essence of the acoustic Airy-Talbot waves?
 4. In the line 6 of page 4, "to perfectly to reproduce." should be "to perfectly reproduce."
 5. In the line 13 of page 4, "in ref. [43]" should be "in Ref. [43]".
 6. In the line 3 (from the bottom) of page 8 in the Supplementary, "the one above the metasurface maintain" should be "the one above the metasurface maintains".
- In my opinion, the manuscript is innovative and could be published if the authors will be better addressing the above issues.

Reviewer #3 (Remarks to the Author):

Report on NCOMMS-23-12315:

The authors investigated the superwavelength self-healing of spoof surface sonic Airy-Talbot waves. They combine the Talbot effect with Airy beams using an acoustic setting and showcase their remarkable resilience against large rigid object, during which sonic self-healing persists. Finally, the authors judiciously phase-and-amplitude launch sound above the structured surface and experimentally demonstrate how the scattered fingerprints of the Airy-Talbot effect remain largely unaffected against superwavelength obstacles, ideal for acoustic functional devices enabling robust transport of encoded signals. Such results may have potential applications in self-healing or diffraction-free information processing devices.

Self-imaging of waves for nonperiodic waves along a parabolic trajectory encompasses both the Talbot effect and accelerating Airy beams. The manuscript reports a significant advance and offers incremental improvement to existing work in the references. The similar dual accelerating Airy-Talbot recurrence effect [Opt. Lett. 40, 5742 (2015)] are studied theoretically. The present manuscript might be of interest to a specific community and not of the general audience. The authors should clarify and discuss the broad interesting and significant progress of the superwavelength self-healing of spoof surface sonic Airy-Talbot waves. In addition, the English throughout this manuscript should be further revised to meet the standard of Nature Communications. The authors should give a brief discussion on this issue in the introduction section to highlight the significant novelty and broad impact of the current work.

The author's work of the superwavelength self-healing of spoof surface sonic Airy-Talbot waves (studied experimentally) is interesting. Comprising Airy beams with self-healing features, such result is ideal in the use for non-topological resilient data transfer of parallel binary signals and may have far-reaching implications in diverse fields ranging from high-capacity communication to on-chip applications. After above improvement, I could recommend publication of the manuscript in Nature Communications.

1. Detailed response to the report of the referees

The report of referee #1:

The authors combine Talbot effect with Airy beam to show that sound waves can remain largely unaffected against large obstacles in the propagating path. The Airy-Talbot waves take full advantage of the self-healing property of Airy beam and the self-imaging property of Talbot effect. This work represents a very good example of how physicists cleverly employ metamaterial-based SSAW that controls sound at deep-subwavelength scale to enable such an important effect. Moreover, they harness the distinct property of Airy-Talbot waves to realize interesting application of multiplexing communication with great potential. All these functionalities are experimentally demonstrated, and the results are well presented. I will recommend this work to be published in Nature Communications if the following issues are well addressed.

Response: We sincerely thank the referee for offering positive remarks on our work and providing valuable suggestions that helped to improve our manuscript. In light of the referee's report, we have performed more numerical simulations and experimental measurements and made every effort to carefully revise the manuscript. We have given the response individually to each comment below.

Comment 1: All the phenomena are studied in holey metasurface that sustains SSAWs. A plenty of simulations are conducted to compare the Airy-Talbot effect in this circumstance with that in air. Some detailed discussion should be given to explain why the metasurface holds better performances.

Response: We thank the referee for pointing out this important issue. In the supplementary materials, we showcase the great capabilities of metasurfaces, through the plots of Figs. S2, S4 and S6, of achieving subwavelength transmission of sound waves, effective compression of sound source size and flexible control of self-imaging distance when compared to the conditions in air. Following the referee's suggestion, we have provided a comprehensive explanation of the enhanced performance brought by metasurfaces and added the detailed discussion in Supplementary Notes 2, 4 and 5.

In Fig. S2, we plot the amplitude distribution curves at the first self-imaging plane, which obviously shows a narrower full width at half maximum (FWAH) of SSAW-based main lobe than that in free space, demonstrating the property of subwavelength propagation brought by the metasurface. This feature can also be explained by

calculating the characteristic propagation distance of Airy beams, within which the spatial FWHM width of the main lobe can remain almost invariant. It has been theoretically proved the characteristic propagation distance of finite energy Airy beams can be expressed as follows [Opt. Commun. 414, 5-9 (2018)]:

$$x_{\max} = 2kb^{-3/2} \sqrt{|y_{\text{cut}}| - |a_1|/b}, \quad (\text{R1})$$

where y_{cut} and a_1 represent the position coordinate of the truncation point and the position of the peak of main lobe at the onset of propagation, respectively. k is the wavenumber and $b=15 \text{ m}^{-1}$ is the scale factor. From Eq. (R1), we find the characteristic distance depends on several factors, including the beam's initial conditions, such as its width and curvature, as well as the wavelength of the acoustic waves. When $a_1 = 0$, Equation (R1) can be simplified as:

$$x_{\max} = 2kb^{-3/2} \sqrt{|y_{\text{cut}}|}. \quad (\text{R2})$$

Under the same truncation length and operation frequency, the characteristic propagation distance of Airy beams under these two circumstances is solely determined by the wavenumber k . At the working frequency of 3170Hz, the propagation wavenumber of sound waves in free space is $k_0 = 58\text{m}^{-1}$. After substituting relevant parameters into the dispersion relationship of supported evanescent wave mode (Eq. (1) in our manuscript), we calculate the wavenumber of SSAWs above the metasurface as $k_{\text{SSAW}} = 78.5\text{m}^{-1}$, which is obviously larger than that in free space. We also plot in Fig. R1(a) the dependence of the maximum nondiffracting propagation distance on the position of the truncated-point y_{cut} , from which we can clearly find the x_{\max} of SSAWs is always larger than that in free space under the same source scale. Specifically, we set $y_{\text{cut}} = 0.15\text{m}$ and substitute k_0 and k_{SSAW} into Eq. (R2) to calculate the characteristic propagation distances in two circumstances: $x_{\max-\text{air}} \approx 0.7\text{m}$ and $x_{\max-\text{SSAW}} \approx 1\text{m}$. This suggests that for a specific observation position close to $z_{\max-\text{air}}$ (e.g., $x=0.65\text{m}$), the beam in free space starts to diffract and spread out, while the main and side lobes of SSAW-based Airy beam remain almost invariant due to the longer characteristic distance $z_{\max-\text{SSAW}}$ (see as Fig. S2), demonstrating the significant advantages of SSAWs in terms of subwavelength transmission. Notice that our

mechanism is general and the performance parameters can also be designed according to the practical requirements.

In Fig. S4, we provide the comparison of source profiles, which is in free space and above the SSAW metasurface respectively, and find by delicately engineering the unit cell of the metasurface, the size of the sound source can be effectively compressed. In Ref. 43 in our manuscript, the transverse scale of the light source is much longer than the working wavelength. Considering the long wavelength of acoustic waves, the source size becomes inevitably large, which is unpractical for the experimental implementation of the acoustic Airy-Talbot effect. Since the distance between two neighboring peaks in the transverse intensity profile is limited to the wavelength of the acoustic Airy beams used [Phys. Rev. A 89, 023807 (2014)], by utilizing the metasurface-based SSAWs, whose wavelength can be effectively compressed to acquire higher spatial resolution, the diffraction limit can be broken and the compactness of the setup can be increased effectively, which indeed solves a crucial problem of experimental implementation (Fig. S3). Furthermore, the realization of acoustic Airy-Talbot effect beyond diffraction limit opens up new avenues for metamaterial-based high-capacity communication paradigm compatible with the conventional multiplexing mechanisms, which we believe has important applications within these established areas and will mark a major step towards the existing mechanisms which only involve the realization of self-accelerating beams [e.g., Phys. Rev. Lett. 107, 116802 (2011)/ Phys. Rev. Lett. 113, 123902 (2014)].

In Fig. S6, we demonstrate the flexible modulation of Talbot distance with the metasurface and ensure the observation of Airy-Talbot effect at a farther distance. According to Eq. (S9) in the supplementary material, we conclude that self-imaging distance is inversely proportional to the wavelength and we can increase or reduce the equivalent wavelength by designing the unit cells of the metasurface to modulate the Talbot distance, which is distinguished from the conventional Talbot effect using the separation of the Airy beams. To verify this, we plot two acoustic fields above the metasurfaces with different structural parameters, as shown in Figs. R1(b) and (c). According to the dispersion relation, we can calculate the wave vectors of SSAWs under these two circumstances as 60m^{-1} and 80m^{-1} , respectively. According to Eq. (S9), we theoretically calculate the self-imaging distances approximately as 0.5m and 0.8m under the same source distribution, which are consistent with the simulated results, demonstrating the capability of metasurfaces in modulating Talbot distances. In addition, the sound energy is mainly concentrated above the metasurface via the local-

resonance-controlled SSAWs and does not leak into free space, which opens up possibilities for manipulating the wave characteristics for specific applications, such as waveguiding. Furthermore, due to the larger wave vector and the longer characteristic propagation distance of metasurface-based Airy beams mentioned above, we are able to suppress the diffraction of Airy beams by structuring subwavelength resonators, which enables the observation of the self-imaging effect at an extended distance, as shown in Fig. S4(b). Based on the referee's comment, more detailed discussion has been added in Supplementary Notes 2, 3 and 5.

FIG. R1. (a) The dependence of the maximum nondiffracting propagation distance of an aperture-truncated Airy beam on the position of the truncated-point y_{cut} . The red and blue solid lines represent the SSAW and the sound waves in free space. Sound field plots of the Airy-Talbot effect with different wavenumber (b) $k_{\text{eff}} = 60\text{m}^{-1}$ (the diameter of aperture D of the resonator is 0.3cm) and (c) $k_{\text{eff}} = 100\text{m}^{-1}$ (the diameter of aperture D of the resonator is 0.64 cm). The first Talbot distance is marked by the red dotted lines in (b) and (c).

Comment 2: In Fig.2, the simulated acoustic fields for different cylinders are well presented. For the measured results, only the line plots are given. The according acoustic field distributions should also be measured to make the results more convincing.

Response: We thank the referee for this suggestion. In the original version of manuscript, we only measured the amplitude distribution at the self-imaging plane (Figs.2 and S7), which is not sufficient. Following the referee's valuable suggestion, we have measured the according acoustic field distributions and more detailed discussion has been added in Supplementary Note 12.

In order to demonstrate the self-healing feature of Airy-Talbot effect better, the acoustic field distributions have been measured, as shown in Fig. R2. The experimental measurement area is $0.4\text{m} \times 0.8\text{m}$, which is denoted by the red box in the simulated

acoustic field. The entire sound field near the metasurface sample is measured utilizing a 1/4-in. free-field microphones (Brüel & Kæjr type-4961) which is attached to a 3D stepping motor to scan the target region point by point. The measured height is selected as 2cm above the metasurface sample for all measurements in the x - y plane. The experimental results demonstrate that the sound field distribution remains almost unchanged despite the presence of the cylinder scatters. In particular, the pronounced pressure peak of the main lobe at the center ($y=0.43$ m) shows the remarkable self-healing property even when the radius of the obstacle reaches $ka=8.8$ (Fig. R2(k)), which reveals the high-quality robust data transmission and would be highly desired for diverse fields ranging from underwater to on-chip communications.

FIG. R2. Experimental demonstration of self-healing feature of Airy-Talbot effect. Scatterers with different radii((a) $ka=1$, $a=1.3$ cm (b) $ka=2$, $a=2.5$ cm (c) $ka=4$, $a=5.1$ cm (g) $ka=6$, $a=7.6$ cm (h) $ka=8.8$, $a=11.2$ cm (i) $ka=10$, $a=12.7$ cm) are located at ($x=0.15$ m, $y=0.6$ m), respectively. k is the equivalent wave number and a is the radius of the obstacle. (d-f) and (j-l) Experimentally measured amplitude fields for the area marked within the red box denoted in (a-c) and (g-i).

Comment 3: The self-imaging of the incident source should be reproduced at the periodic intervals of Talbot length, so I wonder what's the acoustic field distribution at the next Talbot length.

Response: We thank the referee for this important question. As pointed out by the referee, the self-imaging of the incident source should be reproduced at the periodic intervals of Talbot length. We have validated this conclusion through simulation, as shown in Fig. R3(a), which indicates that the sound energy flow will realize a regular replication when we superimpose the infinite energy Airy beams. However, the corresponding experiment will need an extremely large setup and complicate the design of the sound source. In order to facilitate the experimental implementation, we use a 1-metre active phased array to emit two Airy beams to observe this self-imaging effect, resulting in the acoustic field reproduced for finite times. We would like to emphasize that by further expanding the experimental space and using loudspeakers or metasurface with smaller scales, we should be able to observe the self-imaging effects for multiple times.

To verify this point, we depict the sound amplitude plots of the Airy-Talbot effect using sound sources of different lengths to show the influence of the source's length on the Airy-Talbot effect, as shown in Figs. R3(b)-(c), from which we can find as the length decreases, the value of the characteristic propagation distance x_{\max} becomes smaller, resulting in lower periodic repetitions. To be specific, although the amplitude profiles at two self-imaging planes are similar in our current experiment, the central peak at the second Talbot distance gradually shifts from its original position and the value becomes smaller than the lobe's on its left, as shown in Figs. R3(e) and (f), revealing that the self-imaging effect can only take place for a few times and will disappear with the propagation of SSAWs. However, by extending the length of the source, the periodic repetition could happen for more times.

FIG. R3. Pressure amplitude plots of transmitting (a) two infinite energy Airy beams and finite energy Airy beams with (b) the position of the truncated-point $y_{\text{cut}}=1.5\text{m}$ and (c) $y_{\text{cut}}=1\text{m}$, where the first and second Talbot distances are marked by the red dotted lines. (d)-(f) Pressure amplitude plots and profiles at the sound source and two self-imaging planes denoted by red dotted lines in (a)-(c), respectively.

Comment 4: In my opinion, more incident Airy beams will allow a better Talbot effect and more communication channels, but why only two Airy beams are used in this work?

Response: We thank the referee for this important question. As the referee mentioned, more incident Airy beams will allow a better Talbot effect and more communication channels, which, however, will call for a larger experimental space and more complicated design of multi-drivers. In our current design, in order to facilitate the experimental implementation, we emit two Airy beams to display the Airy-Talbot effect, which is sufficient to validate the mechanism of this self-imaging effect and can significantly reduce the size of the sound source. More detailed discussion is explained in the following:

Firstly, by superimposing the Airy beams which satisfy the spatial distribution of Airy beams described by Eq. (3) in the manuscript, the amplitude distribution of the sound field at the first self-imaging plane and the sound source can be written as:

$$\begin{aligned} \left| \phi\left(\frac{4\pi}{\Delta}, y\right) \right| &= \left| \sum_n c_n \text{Ai} \left(y - \left(\frac{2\pi}{\Delta} \right)^2 - n\Delta \right) \right|, \\ \left| \phi(0, y) \right| &= \left| \sum_n c_n \text{Ai} (y - n\Delta) \right|, \end{aligned} \quad (\text{R3})$$

from which we can find the Airy-Talbot effect needs the Airy beams to maintain the acceleration-free wave packet, while the conventional Talbot effect requires the incident waves to be periodic.

Secondly, Figures R4(d-f) show that as more Airy beams are superimposed, the amplitude profile with more subwavelength details appears where the narrowest width of each lobe on the right is nearly 3cm, making the desired discrete amplitude profile hard to realize with the 1.3-in. loudspeakers used in our experiment. Considering this technical issue, we used only two Airy beams to explore this effect.

Last but not least, we depict the sound amplitude plots of the Airy-Talbot effect by the superposition of 2, 3 and 10 Airy beams, as shown in Figs. R4(a-c), from which we find more communication channels can be built when we set the launch points at $y=0\text{m}$, $y=0.45\text{m}$, $y=0.9\text{m}$, etc. However, considering the solutions of the Airy function change from oscillatory to exponential attenuation at the turning point, the n th Airy beams ($n>2$) hardly affect the interference sound field of the first two Airy beams. Figures R4(d-f) show the amplitude profiles of the first self-imaging plane, as marked by red dotted lines in (a-c), from which we can find the amplitude profile by the superposition of the first two Airy beams ($y<0.8\text{m}$) maintains almost unchanged in all three conditions, implying that we can explore this effect with two Airy beams, which is adequate to showcase the Airy-Talbot mechanism.

To conclude, to set up the experimental implementation, we selected two Airy beams to explore this self-imaging effect. Both theoretical and experimental results have sufficiently demonstrated the mechanism. As pointed out by the referee, more incident Airy beams will allow a better Talbot effect, which however requires larger experimental space and should certainly be investigated in the near future. In addition, we have also quantitatively estimated the influence of the thermoviscous losses on the Airy-Talbot effect. We calculate the equivalent refractive index at 3170Hz nearly as $1.42 - 0.02i$, which shows that the inherent losses and wave distortion can be considered negligible in this problem.

FIG. R4. Pressure amplitude plots of transmitting (a) $n=2$, (b) $n=3$ and (c) $n=10$ Airy beams. The upper-right inset shows the zoomed-in view of the black box in (c). Pressure amplitude profiles at the first self-imaging plane of transmitting (d) $n=2$, (e) $n=3$, and (f) $n=10$ Airy beams, which are marked by the red dotted lines in (a)-(c).

The report of referee #2:

In this manuscript, the authors study the self-imaging of waves for nonperiodic waves along a parabolic trajectory encompassing both the Talbot effect and accelerating Airy beams. Beyond the ability to guide waves along a bent trajectory, the self-imaging component adds invaluable advantages towards lensless imaging comprising periodic repetition of planar field distributions. In order to circumvent adverse effect of viscothermal losses and diffractive spread, they structure subwavelength resonators in an acoustically impenetrable surface sustaining spoof surface acoustic waves (SSAWs) to provide highly confined Airy-Talbot effect. Moreover, these spoof surface sonic Airy-Talbot beams that entirely are controlled by geometrical means, enable both extended Talbot distances and compressed subwavelength lobes in the perpendicular direction. From a linear array of loudspeakers, we judiciously phase-and-amplitude launch sound above the structured surface and demonstrate how the scattered fingerprints of the Airy-Talbot effect remain largely unaffected against superwavelength obstacles, ideal for acoustic functional devices enabling robust transport of encoded signals. The topic is interesting. However, some questions and details should be noted and modified by the author. In my opinion, the manuscript is innovative and could be published if the authors will be better addressing the above issues.

Response: We are grateful to the referee for evaluating our paper and we thank the referee for the positive comments and valuable suggestions. We have given the response individually to each comment below.

Comment 1: What is the viscothermal losses?

Response: We thank the referee for this important question. In the Helmholtz resonators (HRs) that we designed, the viscothermal losses typically occur in the narrow regions, which cause attenuation of the SSAWs. To evaluate the influence of the losses on the SSAWs, we calculate the SSAW dispersion relation for a Helmholtz cavity with a narrow neck $r = 0.1 \times 10^{-2} \text{ m}$ [J. Acoust. Soc. Am., 145, 1 (2019)]:

$$\frac{i\omega\rho_0}{\sqrt{k_x^2 - k_0^2}} = \frac{a^2}{\pi(D/2)^2} \left(\rho_0 c_0 \frac{2i \sin(k_c d/2)}{\sqrt{(\gamma - (\gamma - 1)\Psi_h)\Psi_v}} + \frac{i\pi(D/2)^2}{\omega C_{\text{HR}}} \right), \quad (\text{R4})$$

where k_x is the wavenumber of the SSAWs, k_0 is the wavenumber in the free space, ω is the angular frequency, $\rho_0 = 1.21 \text{ kg/m}^3$ is the density of air, $c_0 = 343 \text{ m/s}$ is the

sound speed in air, C_{HR} is the acoustic mass and the acoustic capacitance of HRs, $\mu = 1.85 \times 10^{-5} \text{ kg}/(\text{m}\cdot\text{s})$ is the dynamic viscosity of air, $\kappa = 0.0258 \text{ W}/(\text{m}\cdot\text{K})$ is the fluid thermal conductivity, $c_{p0} = 1.005 \times 10^3 \text{ J}/(\text{kg}\cdot\text{K})$ is the specific heat at constant pressure and $\gamma = 1.4$ is the ratio of the specific heat at constant pressure and constant volume. We introduce the viscous wave number k_v and the thermal wave number k_h as:

$$k_v^2 = -i\omega \frac{\rho_0}{\mu}, \quad k_h^2 = -i\omega \frac{\rho_0 c_{p0}}{\kappa}. \quad (\text{R5})$$

Further, the function of viscous and thermal fields can be, respectively, expressed as:

$$\Psi_v = \frac{J_2(k_v D/2)}{J_0(k_v D/2)}, \quad \Psi_h = \frac{J_2(k_h D/2)}{J_0(k_h D/2)}, \quad (\text{R6})$$

where J_n is the Bessel function of the first kind and order n . Then the complex wave number can be calculated as:

$$k_c^2 = k_0^2 \left(\frac{\gamma - (\gamma - 1)\Psi_h}{\Psi_v} \right). \quad (\text{R7})$$

We have calculated in Fig. R5 the dispersion curve, which illustrates that the real part of the wavenumber of the SSAWs gradually deviates from the lossless dispersion curve while the imaginary part progressively increases as the frequency rises, showing an obvious attenuation along the wave propagation direction. Consequently, we choose a larger diameter of the aperture $D = 0.7\text{cm}$ and a relatively low operating frequency $f = 3170\text{Hz}$ to alleviate the viscothermal effects. Actually, since the thicknesses of the viscous and thermal boundary layers at the operation frequency can be calculated as:

$$d_\mu = \sqrt{\frac{2\mu}{\omega\rho_0}} \approx 3.9 \times 10^{-5} \text{ m}, \quad d_\kappa = \sqrt{\frac{2\kappa}{\rho_0 c_{p0} \omega}} \approx 4.6 \times 10^{-5} \text{ m}, \quad (\text{R8})$$

whose value is much smaller than the radius of the neck, the attenuation of SSAWs caused by the viscothermal losses can be restricted to a relatively low level [Fundamentals of Acoustics, 4th ed. (J. Wiley& Sons, Inc., New York, 2000)], which ensures the self-imaging effect even over an extended distance. To be specific, we calculate the equivalent refractive index nearly as $1.42 - 0.02i$, which shows that the inherent losses only slightly change the index and the attenuation is negligible at the operation frequency.

Based on the referee's comments, more discussions about the effect of the inherent thermal and viscous losses inside the HRs have been added in Supplementary Note 9.

FIG. R5. Calculated dispersion relation of the SSAWs in the presence of the thermal and viscous losses. For the lossy case, the wave vector of the SSAWs becomes a complex number (red solid line: real part; red dashed line: imaginary part). The curves of the lossless case (blue solid line) and the air line (black solid line) are also presented for comparison. Here $f_0 = \sqrt{1/M_{\text{HR}}C_{\text{HR}}}/2\pi$ is the cavity resonance frequency. The unit cell is designed to have $a = 2 \times 10^{-2} \text{ m}$, $t = 0.75 \times 10^{-2} \text{ m}$, $D = 0.2 \times 10^{-2} \text{ m}$ and $d = 0.1 \times 10^{-2} \text{ m}$.

Comment 2: What is the difference between the acoustic Airy-Talbot waves and the Airy-Talbot plasmon?

Response: We thank the referee for this important question. Acoustic Airy-Talbot waves and Airy-Talbot plasmons are two distinct phenomena that occur in different physical systems. In the following, we will briefly describe the differences between them:

It has been theoretically proved that Airy-Talbot plasmons are a particular class of SPP waves that propagates at the interface between a metal and a dielectric material (air) and causes a strong interference along curved trajectories, whose propagation

properties provide potential value for nanoscale plasmonic devices [Opt Lett 47, 7 (2022)]. It is worthwhile to note that due to the high losses of strongly confined surface plasmons, the limited propagation length of Airy beam makes it difficult for the observation of the self-imaging effect, which also needs to be considered in acoustics. We have included the article in the reference list.

Due to the lack of a direct acoustical analog of a “plasmon”, we developed an effective method to investigate the acoustic Airy-Talbot effect along the metasurface supporting SSAWs. By delicately designing the unit cell of the metasurface, we can significantly reduce the impact of thermoviscous losses and realize this effect with high spatial resolution. To the best of our knowledge, the Airy-Talbot plasmon has not been demonstrated in experiment while our work is the first theoretical and experimental realization of Airy-Talbot effect in acoustics. Through the experimental demonstration, we find these waves exhibit a unique property known as “self-healing” or “self-reconstruction”, meaning that they can reform their shape after encountering an obstacle, which has also not been investigated in the previous works. Besides, considering the inherent problems pertaining to acoustic systems, such as limited propagation length, severe diffraction effects and bulky source scale, by introducing the concept of metasurfaces, we enable modulating the self-imaging distance and suppressing the diffraction effect simultaneously, which indeed enables a practical observation of the Airy-Talbot effect in acoustics. Furthermore, we also introduce a new mechanism utilizing subwavelength Airy beam as a novel encoding/decoding degree of freedom in two dimensions while maintaining the reduction of the system dimensions without compromising the high-quality robust data transmission.

In summary, both acoustic Airy-Talbot waves and Airy-Talbot plasmons exhibit self-imaging properties, while they arise in different physical domains and need to be realized with different methods. We would like to emphasize that our manuscript reports the first theoretical and experimental realization of Airy-Talbot effect in acoustic regime, thanks to the introduction of the metasurface. We quantitatively evaluate the self-healing performance of Airy-Talbot effect for the first time and demonstrate its advantages over the traditional Talbot effect in terms of the high robustness against the superwavelength obstacles, which may have potential applications in the high-quality acoustic communications. Based on the referee’s comments, a brief discussion has been added in the Introduction section to highlight the novelty and significance of our work.

Comment 3: What is the physical essence of the acoustic Airy-Talbot waves?

Response: We thank the referee for this important question. Acoustic Airy-Talbot effect is a new type of self-imaging phenomenon that arises from the diffraction and interference of acoustic waves, which can realize the self-imaging of the nonperiodic signals along a curved trajectory and may open up new avenues within diverse important established areas such as the underwater and on-chip communications. However, considering the inherent problems mentioned below, we cannot observe the self-imaging effect directly in free space:

1. Limited propagation length: Due to the severe dissipation (viscous and thermal losses) of acoustic waves (the attenuation coefficient is $\alpha_{\text{acoustic}} \approx 0.0012$ dB/m in air at 3170Hz (operation frequency in our manuscript) [Sensors 18, 499 (2018)], which is much larger than $\alpha_{\text{optics}} \approx 0.0003$ dB/m for optical waves (532nm laser, used in Ref. 43) [Appl. Opt. 12, 896 (1973)]), the effective propagation length of acoustic Airy beams is not sufficient for self-imaging in free space, as shown in Figs. R6(a) and (b). 2. Severe diffraction effects: Caused by diffraction in free space, the Airy beam will quickly spread as it travels farther. In addition, due to the diffraction limit, the evanescent component with large spatial momentum has difficulties to propagate in free space, which makes the amplitude profiles at the subwavelength scale impossible to perfectly reproduce in free space (Figs. R6(c) and (d)). In summary, exploring the Airy-Talbot analogy for acoustics is far from trivial but, respectively spoken, has been a very challenging undertaking.

In our manuscript, we demonstrate this acoustic self-imaging effect for the first time by emitting the metasurface-based SSAWs. Thanks to the unique properties brought by the subwavelength metasurface, the experimental observation with high spatial resolution is enabled (Figs. R6(e)), whose advantages can be concluded as follows:

Firstly, we can flexibly control the SSAWs by engineering the structural parameters of the metasurface, which enables both extended Talbot distances along the propagation direction and compressed subwavelength lobes in the perpendicular direction. Considering the SSAWs, whose wavelength is shorter than that in free space, the diffraction limit can be broken and Airy beams are allowed to propagate at the subwavelength scales. On the other hand, the Talbot distance in our system will increase as governed by the Eq. (S9), which is also visually depicted in Fig. R6(d) (the self-imaging effect is more obvious and the Talbot distance is extended compared with that in free space), suggesting that the sound information carried by subwavelength waves

can transmit through a longer distance compared to free space. By designing the unit cells of the metasurface, therefore, we are able to modulate the self-imaging distance and suppress the diffraction effect simultaneously, which enables Airy beams to interfere adequately at an extended distance, which indeed solves a critical problem of the observation of Airy-Talbot effect in acoustics.

Furthermore, in order to display the wide applications in many areas such as communication, we explore the high-speed communication based on the Airy-Talbot effect in a two-dimensional acoustic system. In comparison with the existing spatial multiplexing sound communication technologies usually demonstrated in free space with severe diffraction and dissipation effects (e.g., acoustic orbital angular momentum communication [Nat. Commun.13, 5171 (2022); PNAS, 114, 7250-7253 (2017)]), our proposed scheme of using subwavelength Airy beam as a new coding/decoding degree of freedom for high-density acoustic communication in two dimensions enables the reduction on the system dimension while keeping a good communication performance. To be specific, thanks to the self-healing and non-diffraction characteristics, an extremely high transmission efficiency and an ultra-low low crosstalk are achieved, which may have far-reaching implications in diverse fields ranging from on-chip signal transfer and filtering to acoustofluidic manipulation. Besides, our designed metasurface also serves as an important platform for exploring wave physics like conformal transformation [PRL 119, 033902 (2017)] which may also find applications such as in digital coding communications.

In summary, by introducing the concept of metasurface, we realize the acoustic Airy-Talbot effect for the first time, which enables surface confined Airy beams along curved trajectories at deeply subwavelength scales and facilitates distinct self-imaging and self-healing capabilities against superwavelength object. Furthermore, we propose a new application using multipath sound transmission to realize real-time high-capacity communication free of signal-processing or sensor-scanning, which has important inspiration for the metamaterial-based high-capacity communication paradigm compatible with the conventional multiplexing mechanisms.

Based on the referee's comments, more details about the physical essence of Airy-Talbot effect have been added in the Introduction section.

FIG. R6. Pressure amplitude plots of transmitting one Airy beam in (a) free space and (b) SSAW metasurface. Sound field plots of the Airy-Talbot effect in (c) free space and (d) SSAW metasurface. The Talbot distance is marked by the red dotted lines in (c) and (d). The simulation is carried out by using the method of effective parameter. (e) Normalized amplitude distributions along $x = 0.65$ m, as marked by the red dotted lines in (a) and (b). (f) Comparison between the diffractionless property of Airy beams in (a) free space and on (b) SSAW metasurface, respectively. The black dotted and solid lines represent the SSAW and the sound waves in free space.

Comment 4: In the line 6 of page 4, “to perfectly to reproduce.” should be “to perfectly reproduce.”

Response: We thank the referee for this comment. We have replaced the expression of “to perfectly to reproduce” with “to perfectly reproduce” on Page 4.

Comment 5: In the line 13 of page 4, “in ref. [43]” should be “in Ref. [43]”.

Response: We thank the referee for this comment. We have replaced the expression of “in ref. [43]” with “in Ref. [43]” on Page 4.

Comment 6: In the line 3 (from the bottom) of page 8 in the Supplementary, “the one above the metasurface maintain” should be “the one above the metasurface maintains” .

Response: We thank the referee for this comment. We have replaced the expression of “the one above the metasurface maintain” with “the one above the metasurface maintains” in Supplementary Note 6.

The report of referee #3:

The authors investigated the superwavelength self-healing of spoof surface sonic Airy-Talbot waves. They combine the Talbot effect with Airy beams using an acoustic setting and showcase their remarkable resilience against large rigid object, during which sonic self-healing persists. Finally, the authors judiciously phase-and-amplitude launch sound above the structured surface and experimentally demonstrate how the scattered fingerprints of the Airy-Talbot effect remain largely unaffected against superwavelength obstacles, ideal for acoustic functional devices enabling robust transport of encoded signals. Such results may have potential applications in self-healing or diffraction-free information processing devices. The author's work of the superwavelength self-healing of spoof surface sonic Airy-Talbot waves (studied experimentally) is interesting. Comprising Airy beams with self-healing features, such result is ideal in the use for non-topological resilient data transfer of parallel binary signals and may have far-reaching implications in diverse fields ranging from high-capacity communication to on-chip applications. After above improvement, I could recommend publication of the manuscript in Nature Communications.

Response: We sincerely appreciate the referee for the positive remarks and the valuable suggestions that help us to improve the manuscript. In light of the referee's report, we have made every effort to carefully revise the manuscript and hope this could address the concerns of referee. We have given the response individually to each comment below.

Comment 1: Self-imaging of waves for nonperiodic waves along a parabolic trajectory encompasses both the Talbot effect and accelerating Airy beams. The manuscript reports a significant advance and offers incremental improvement to existing work in the references. The similar dual accelerating Airy-Talbot recurrence effect [Opt. Lett. 40, 5742 (2015)] are studied theoretically. The present manuscript might be of interest to a specific community and not of the general audience. The authors should clarify and discuss the broad interesting and significant progress of the superwavelength self-healing of spoof surface sonic Airy-Talbot waves.

Response: We thank the referee for raising this question. The Airy-Talbot effect is a new type of self-imaging phenomenon, whose acoustic field is generally not periodic and is self-imaged along curved trajectories, while the acoustic demonstration and corresponding applications have not been explored before. In our manuscript, by utilizing the metasurface-based SSAWs, we observe this acoustic effect experimentally

for the first time. Thanks to the enhanced self-healing property brought forward by the Airy-Talbot effect, which has also not been discussed in any wave system, we propose a brand-new application using multipath sound transmission to realize real-time high-capacity communications in a two-dimensional acoustic system, which is of academic value and practical significance. Besides, the dual accelerating Airy-Talbot recurrence effect mentioned by the referee [realized theoretically and numerically in Opt. Lett. 40, 5742 (2015), which has been added to our reference list] is a good example for demonstrating the effectiveness of our mechanism. Via numerical simulation, we find this self-imaging effect can be observed at the half-Talbot distance on our metasurface (as shown in Figs. R1(a) and (b)) but would otherwise be absent in air. Furthermore, by implementing the metasurface-based scheme proposed in our manuscript, we may be able to experimentally observe this special effect, underscoring the potential interest and significant progress in our work. Following the referee's suggestion, we have added a brief discussion in the Introduction section to highlight the significance of our work.

FIG. R7. (a) Pressure amplitude plots of dual Airy-Talbot effect with $c_n = [\dots 1, i, 1, i, \dots]$, where the first half-Talbot and Talbot distances are marked by the red dotted lines. (b) Pressure amplitude profiles at the sound source, half-Talbot and Talbot distances denoted by red dotted lines in (a). (c) Comparison between the input and output signals with a cylinder ($ka=8.8$ with a sound-soft boundary) placed at $(x=0.15\text{m}, y=0-0.8\text{m})$. Cross correlation describes the similarity between the initial beam's field and propagating beam's field.

In the following, we will discuss the novelty of our work in details. Firstly, we would like to illustrate the inherent problems pertaining to sound waves physics that challenges the direct realization of Airy-Talbot effect in free space:

1. Limited propagation length: Due to the severe dissipation (viscous and thermal

losses) of acoustic waves (the attenuation coefficient is $\alpha_{\text{acoustic}} \approx 0.0012$ dB/m in air at 3170Hz [Sensors 18, 499 (2018)], which is much larger than $\alpha_{\text{optics}} \approx 0.0003$ dB/m for optical waves (532nm laser, used in Ref. 43) [Appl. Opt. 12, 896 (1973)]), the effective propagation length of acoustic Airy beams is not sufficient for self-imaging in free space, as shown in Figs. R8(a) and (b). **2. Severe diffraction effects:** Caused by diffraction in free space, the Airy beam will quickly spread as it travels farther. In addition, considering the fact that evanescent component with large spatial momentum has difficulties to leak into free-space, it is impossible for the complex amplitude profiles, which is superimposed by two Airy beams, to perfectly reproduce in free space (Figs. R8(c) and (d)). **In summary, exploring the Airy-Talbot analogy for acoustics is far from trivial but, respectively spoken, has been a very challenging undertaking.**

Considering the inherent problems mentioned above, by introducing the concept of metasurfaces, we enable modulating the self-imaging distance and suppressing the diffraction effect at a compact scale, which indeed **solves a critical problem concerning the observation of the Airy-Talbot effect in acoustics**. Furthermore, by implementing the metasurface-based platform, we may be able to modulate the effective parameters and design the metasurface with gradient refractive index profiles [Research, 1748537 (2019)], which has important inspiration for researches on acoustic functional devices enabling robust transport of encoded signals and can be investigated in our future work.

In addition, although self-healing effects of Airy beam and Talbot effect have both been investigated before, the non-diffraction property of Airy-Talbot effect, which is the combination of these two concepts, has also not been studied yet. Our work has **quantitatively shown the self-healing performance of Airy-Talbot effect for the first time and demonstrated its advantages compared to the traditional Talbot effect, which may have potential applications in robust acoustic communications**.

Here we would like to add that more experimental measurements of acoustic field distributions for self-healing feature have been updated in Supplementary Note 9.

Furthermore, we **introduce a new mechanism utilizing subwavelength Airy beam as a novel encoding/decoding degree of freedom in two dimensions** while maintaining the reduction of the system dimensions without compromising the high-quality robust data transmission. We believe that our findings may open a new avenue within diverse important established areas such as underwater [e.g., Phys. Rev. Lett.

117, 097403 (2016)] and on-chip communications [e.g., Lab on a Chip 15, 2722 (2015)], which otherwise suffer from the complexity of the propagating media due to the existence of randomly distributed bubbles or modules. To be specific, in practical circumstances wherein the location of obstacles is random and changing in real time, obstacle-induced scattering effect is difficult to suppress since we do not know the exact region needed to be controlled. However, our proposed method, which utilizes the self-healing property of Airy beams, is robust against the position and property of scatters (i.e., rigid or soft). To inspect such robustness of our mechanism, we change the position of a soft scatter along the y axis over the transmission region and numerically calculate the cross-correlation in Fig. R7(c), which showcases a large cross-correlation coefficient exceeding 0.5 for all of the scatter position, demonstrating a relatively high transmission accuracy regardless of the obstacle's position and property.

In summary, we **demonstrate the acoustic Airy-Talbot effect for the first time** by structuring a subwavelength SSAW metasurface. We have argued that it is impossible to realize the acoustic version of the Airy-Talbot in free-space, both because of inherent thermoviscous losses and serious diffraction. In addition, **we quantitatively evaluate the self-healing performance of Airy-Talbot effect for the first time and demonstrate its advantages compared to the traditional Talbot effect in terms of the robustness against obstacles**, which has not been even explored in wave physics. Furthermore, **we propose a new application using multipath sound transmission to realize real-time high-capacity communication** free of signal-processing or sensor-scanning, which has important inspiration for the metamaterial-based high-capacity communication paradigm compatible with the conventional multiplexing mechanisms. Last but not least, **our mechanism is general and more functional devices in other systems can also be designed according to the practical requirements**. Hence, we believe our work is of general interest and significant progress to the wide audience of Nature Communications.

FIG. R8. Pressure amplitude plots of transmitting one Airy beam in (a) free space and (b) SSAW metasurface. Sound field plots of the Airy-Talbot effect in (c) free space and (d) SSAW metasurface. The Talbot distance is marked by the red dotted lines in (c) and (d). The simulation is carried out by using the method of effective parameter. (e) Normalized amplitude distributions along $x = 0.65$ m, as marked by the red dotted lines in (a) and (b). (f) Comparison between the diffractionless property of Airy beams in (a) free space and on (b) SSAW metasurface, respectively. The black dotted and solid lines represent the SSAW and the sound waves in free space.

Comment 2: In addition, the English throughout this manuscript should be further revised to meet the standard of Nature Communications. The authors should give a brief discussion on this issue in the introduction section to highlight the significant novelty and broad impact of the current work.

Response: We thank the referee for this important suggestion. In the updated version, we have improved the English of our manuscript and added a brief discussion in the Introduction section to highlight the significance of our work.

2. Detailed changes made to the manuscript are the following:

(1) Page 2, line 10: We have emphasized the novelty in the abstract: *“In order to circumvent thermoviscous and diffraction effects, we structure subwavelength resonators in an acoustically impenetrable surface supporting spoof surface acoustic waves (SSAWs) to provide highly confined Airy-Talbot effect, extending Talbot distances along the propagation path and compressing subwavelength lobes in the perpendicular direction. From a linear array of loudspeakers, we judiciously control the amplitude and phase of the SSAWs above the structured surface and quantitatively evaluate the self-healing performance of Airy-Talbot effect by demonstrating how the distinctive scattering patterns remain largely unaffected against superwavelength obstacles. Furthermore, we introduce a new mechanism utilizing subwavelength Airy beam as a coding/decoding degree of freedom for acoustic communication with high information density comprising robust transport of encoded signals.”*

(2) Page 4, line 54: We have added a brief discussion about the difficulty of realizing the Airy-Talbot effect in acoustic regime: *“First, caused by severe diffraction in free space, an acoustic Airy beam undergoes rapid spreading during propagation, which hinders the collective superposition of several Airy beams and perfect periodic reproduction of the subwavelength amplitude profiles. Moreover, the attenuation coefficient of sound waves in free space is significantly larger than that of optical waves, which means that thermoviscous losses restrict the extended propagation lengths and the formation of self-imaging in the acoustic regime. Lastly, considering the long wavelength of acoustic waves, the experimental implementation will be unpractically large in comparison to the optical counterpart⁴⁶. Conclusively, the experimental observation of the acoustic Airy-Talbot effect still remains challenging.”*

(3) Page 4, line 66: We have highlighted the novelty in the Introduction section: *“In this article, we report the first theoretical and experimental realization of the Airy-Talbot effect using sound waves. By employing a metasurface, we combine the Talbot effect with Airy beams and quantitatively evaluate their remarkable resilience against superwavelength rigid object, during which sonic self-healing persists. In order to overcome the above-named problems in terms of attenuation and diffraction, we*

structure Helmholtz resonators (HRs) into an otherwise acoustically impenetrable surface to sustain SSAWs. Our demonstration showcases a significant reduction in the transverse lobe profiles, alongside the capacity to extend the self-imaging of multiple Airy beams along curved paths to farther distances, both thanks to the subwavelength SSAWs. Thus, the realization of surface-confined self-imaging along curved trajectories at deeply subwavelength scales, facilitates distinct self-healing capabilities against superwavelength objects. Furthermore, we experimentally exemplify resilient parallel multiplexing of acoustic images by utilizing subwavelength Airy beam as a novel encoding/decoding degree of freedom in two dimensions. Despite the reduction of system dimensions in comparison with the existing free-space sound communication paradigm, our proposed scheme enables high-quality robust data transmission, which should be significant for diverse applications ranging from on-chip signal transfer⁴⁷ and filtering to acoustofluidic manipulation⁴⁸.”

(4) Page 7, line 122: We have added a discussion about the subwavelength propagation property: *“To further demonstrate the subwavelength propagation property, we compare the amplitude profiles of a single Airy beam propagating in free space and above the metasurface, showing the subwavelength spatial compression above the SSAWs metasurface (see Supplementary Notes 3-5 for more details).”*

(5) Page 7, line 130: We have added a brief description about the modulation of the Talbot distance with the metasurface: *“see Supplementary Note 6 wherein we demonstrate the capability of metasurfaces in modulating Talbot distances.”*

(6) Page 8, line 151: We have added a discussion about the validity of our proposed method by using two Airy beams: *“It is worth noting that more incident Airy beams will allow for a better Talbot effect, which, however, calls for a larger experimental setup and a more complicated design of the sound source. In our current design, to facilitate the experimental implementation, we emit two Airy beams to display the Airy-Talbot effect, which is sufficient to validate the mechanism of the self-imaging effect (See Supplementary Note 8 for more details).”*

(7) Page 8, line 157: We have added a discussion about the thermoviscous losses: *“In addition, for the purpose of avoiding thermal viscosity and anisotropy, we set the*

operation frequency at 3170Hz, which is slightly off-resonance (see Supplementary Note 9 for the discussion of thermoviscous losses)."

(8) Page 9, line 176: We have added a brief description about the experimental setup: *"The measurement is carried out in an anechoic chamber for eliminating the undesired reflections from the boundaries."*

(9) Page 9, line 181: We have added a brief description of the 3D stepping motor: *"The entire sound field near the metasurface sample is measured utilizing a 1/4-in. free-field microphones (Briuel & Kæjr type-4961) which is attached to a 3D stepping motor to scan the target region point by point."*

(10) Page 10, line 193: We have added a discussion about the influence of the source's length on the Airy-Talbot effect: *"Considering the size of our source, the acoustic field can only be reproduced a limited number of times. Yet, extending the source length increases the number of reproductions (see Supplementary Note 10 for more discussions about the influence of the source's length on the Airy-Talbot effect)."*

(11) Page 11, line 217: We have added a brief discussion about the experimental results of the self-healing property in the Airy-Talbot effect: *"see Supplementary Notes 11 and 12 where we demonstrate the robust transmission with scatterers of different radii by measuring the acoustic field."*

(12) Page 12, line 266: We have emphasized the generality of our mechanism in the Discussion section: *"Furthermore, our mechanism is general and enables design of more versatile functional devices in other systems."*

(13) Supplementary Note 3: We have added a detailed discussion about the subwavelength propagation feature of the SSAWs and given some theoretical results to prove that.

(14) Supplementary Note 4: We have added a brief discussion about the comparison of source profiles in free space and on the metasurface.

(15) Supplementary Note 5: We have added a brief discussion about the diffraction-free property of the SSAWs.

(16) Supplementary Note 6: We have added a detailed description about the enhanced performance and the potential applications of the metasurface.

(17) Supplementary Note 8: We have added a detailed discussion about the validity of the proposed method by using two Airy beams. The numbers of the rest Supplementary Notes have been updated accordingly.

(18) Supplementary Note 9: We have added a detailed discussion about the effect of the thermoviscous losses and demonstrated the attenuation can be negligible in our experiment. The numbers of the rest Supplementary Notes have been updated accordingly.

(19) Supplementary Note 10: We have added a detailed discussion about the influence of the source's length on the Airy-Talbot effect. The numbers of the rest Supplementary Notes have been updated accordingly.

(20) Supplementary Note 12: We have added a detailed discussion of the experimental demonstration of self-healing feature of Airy-Talbot effect, which is basically consistent with the simulated results. The numbers of the rest Supplementary Notes have been updated accordingly.

(21) References: We have added a work of the topological acoustics in Ref. 26. We have added a work of the Airy-Talbot plasmon in Ref. 33. We have added a work about realizing the dual accelerating Airy-Talbot recurrence effect in Ref. 34. We have added a work about the on-chip applications in Ref. 47. We have added a work about the experimental realization of topological on-chip acoustic tweezers in Ref. 48. In the supplementary materials, we have added a work about the maximum nondiffracting propagation distance of Airy beams in Ref. 3. We have added a work about the characteristics of the Airy beams in Ref. 4. We have added a work about the thermoviscous losses in the Helmholtz resonators in Ref. 6. We have added a work which introduces the concept of the thermoviscous losses in Ref. 7. The numbers of the rest references have been updated accordingly.

(22) We have improved the English of the manuscript.

Reviewer #1 (Remarks to the Author):

All the comments are well addressed by the authors. I recommend this work to be published in Nature Communications.

Reviewer #2 (Remarks to the Author):

The revised version of this manuscript can be accepted for publication in Nature Communications.

Reviewer #3 (Remarks to the Author):

Now, I recommend publication of the manuscript in Nature Communications.

1. Detailed response to the report of the referees

The report of referee #1:

All the comments are well addressed by the authors. I recommend this work to be published in Nature Communications.

Response: We respectfully thank referee #1 for the valuable and constructive comments, which have helped us in improving our manuscript clarity.

The report of referee #2:

The revised version of this manuscript can be accepted for publication in Nature Communications.

Response: We respectfully thank referee #2 for the efforts of reviewing our manuscript and offering valuable suggestions that helped to improve our manuscript.

The report of referee #3:

Now, I recommend publication of the manuscript in Nature Communications.

Response: We sincerely thank referee #3 for offering positive remarks on our work and valuable suggestions that have helped us to improve our manuscript.

2. Detailed changes made to the manuscript are the following:

- (1) **Page 3, line 31:** We have replaced the expression of “*These remarkable responses and wave functionalities*” with “*These exceptional responses and wave functionalities*”.
- (2) **Page 4, line 66:** We have replaced the expression of “*the first theoretical and experimental realization of the Airy-Talbot effect*” with “*the theoretical and experimental realization of the Airy-Talbot effect*”.
- (3) **Page 4, line 69:** We have replaced the expression of “*evaluate their remarkable resilience against superwavelength rigid object*” with “*evaluate their resilience against superwavelength rigid object*”.
- (4) **Page 12, line 244:** We have replaced the expression of “*results in an extremely high accuracy*” with “*results in a high accuracy*”.
- (5) **Page 12, line 257:** We have rewritten this sentence in the Discussion section “*In conclusion, the Airy-Talbot effect is experimentally demonstrated for sound on a metasurface*”.